# Coincidence detection and bi-directional transmembrane signaling control a bacterial second messenger receptor

Richard B Cooley*[†], John P O'Donnell, Holger Sondermann*

Department of Molecular Medicine, College of Veterinary Medicine, Cornell University, Ithaca, United States

**Abstract** The second messenger c-di-GMP (or cyclic diguanylate) regulates biofilm formation, a physiological adaptation process in bacteria, via a widely conserved signaling node comprising a prototypical transmembrane receptor for c-di-GMP, LapD, and a cognate periplasmic protease, LapG. Previously, we reported a structure-function study of a soluble LapD•LapG complex, which established conformational changes in the receptor that lead to c-di-GMP-dependent protease recruitment (Chatterjee et al., 2014). This work also revealed a basal affinity of c-di-GMP-unbound receptor for LapG, the relevance of which remained enigmatic. Here, we elucidate the structural basis of coincidence detection that relies on both c-di-GMP and LapG binding to LapD for receptor activation. The data indicate that high-affinity for LapG relies on the formation of a receptor dimer-of-dimers, rather than a simple conformational change within dimeric LapD. The proposed mechanism provides a rationale of how external proteins can regulate receptor function and may also apply to c-di-GMP-metabolizing enzymes that are akin to LapD.

*For correspondence: cooleyr@ oregonstate.edu (RBC); hs293@ cornell.edu (HS)

Present address: [†]Department of Biochemistry and Biophysics, Oregon State University, Corvallis, United States

Competing interests: The authors declare that no competing interests exist.

## Introduction

Bacteria exist either as free swimming, planktonic cells or as multi-cellular biofilms — aggregates adhered to biotic or abiotic surfaces, which are enveloped in a self-produced matrix composed of proteins, nucleic acids, and polysaccharides (*Hall-Stoodley et al., 2004*; *Teschler et al., 2015*). This matrix protects bacteria from hostile environments, rendering them more resistant to insult and, in the context of pathogenic microbes, more tolerant to antibiotics and the immune system. The transition between planktonic and sessile lifestyles is tightly regulated, typically involving the second messenger c-di-GMP (or cyclic diguanylate) (*Hengge, 2009*; *Hengge et al., 2016*; *Römling et al., 2005*; *Wolfe and Visick, 2008*). The cellular c-di-GMP level is governed by two enzyme classes with opposing activities: GGDEF domain-containing diguanylate cyclases and EAL or HD-GYP domain-containing phosphodiesterases, which synthesize and degrade the dinucleotide, respectively (*Schirmer and Jenal, 2009*). A wide variety of specific, c-di-GMP-binding receptors translate the second messenger signal into physiological responses (*Chou and Galperin, 2016*; *Sondermann et al., 2012*). Bacterial genomes can encode a large number of c-di-GMP-signaling proteins and this number scales roughly with the organism's adaptivity (*Galperin, 2005*). A particularly interesting observation is that, despite identical catalytic activities, c-di-GMP-metabolizing enzymes often confer distinct signaling outcomes (*Abel et al., 2011*; *Dahlstrom et al., 2015*, *2016*; *Ha et al., 2014*; *Kulasakara et al., 2006*; *Lindenberg et al., 2013*; *Newell et al., 2011a*). The mechanisms underlying this apparent signaling specificity are not well understood.

Recent work by our lab and others have focused on a conserved signaling system comprising the Lap operon that controls cell adhesion and biofilm formation in gammaproteobacteria, including *Pseudomonas fluorescens* (*Chatterjee et al., 2014*; *Navarro et al., 2011*; *Newell et al.,*

*2009*, *2011b*), *P. aeruginosa* (*Cooley et al., 2016 Rybtke et al., 2015*), *P. putida* (*Gjermansen et al., 2010*), *Bordetella bronchiseptica* (*Ambrosis et al., 2016*), and *Shewanella oneidensis* (*Zhou et al., 2015*) (*Figure 1A*). At its center, the inner membrane protein LapD functions as a receptor with degenerate GGDEF and EAL domains, which together relay intracellular c-di-GMP concentrations to the periplasm. At high c-di-GMP levels, LapD sequesters the adhesin protein-specific, periplasmic protease LapG at the inner membrane via the receptor's periplasmic domain. This step ensures that large adhesin proteins whose transcription is activated by the dinucleotide remain stably associated with the outer cell membrane. When c-di-GMP levels drop and adhesin expression ceases, LapD undergoes a conformational change, adopting an autoinhibited state with low affinity for LapG; freed LapG, in turn, processes the adhesin proteolytically, weakening cell adhesion and ultimately contributing to biofilm dispersal (*Chatterjee et al., 2014*; *Navarro et al., 2011*; *Newell et al., 2011b*; *Cooley et al., 2016*; *Rybtke et al., 2015*; *Borlee et al., 2010*; *Martínez-Gil et al., 2014*; *Monds et al., 2007*). Notably, our previous work identified a transient, yet detectable interaction of LapG with c-di-GMP-unbound LapD, suggesting that the protease may participate in an early event of LapD signaling (*Chatterjee et al., 2014*). Curiously, saturation binding of LapG to LapD was markedly lower in the absence of c-di-GMP compared to levels when both ligands, c-di-GMP and LapG, were present. The functional relevance and mechanistic role of this interaction, however, remained poorly defined.

In addition, a subsystem of diguanylate cyclases feeds specifically into the *P. fluoresens* LapD, indicating apparent signaling specificity between enzymes and receptors involved in c-di-GMP signal transduction (*Newell et al., 2011a*). At least one of these enzymes, GcbC, fulfills a distinct role in contributing an activation signal that relies on protein–protein interactions with LapD (*Dahlstrom et al., 2015*, *2016*) (*Figure 1A*). Yet, our previous structural analysis of LapD indicated that a helical motif mediating pairwise interactions with GcbC was occluded in the autoinhibited state (*Figure 1—figure supplement 1*). Furthermore, modeling LapD as a simple dimer also suggested global steric incompatibility between these two transmembrane proteins (*Figure 1A*).

Here, we focus on the molecular basis of switching of the purified, full-length c-di-GMP receptor LapD. These follow-up studies reveal an unanticipated role for LapG, together with c-di-GMP, in establishing the signaling-competent conformation of LapD by inducing higher-order oligomerization of the receptor. On the basis of the results, significant modifications to our model include coincidence detection of dinucleotide and protease as well as bidirectional signaling across the membrane as integral steps in LapD activation, with implications for the origins of transmembrane c-di-GMP signaling and for the regulation of LapD via heterologous interactions with diguanylate cyclases (*Dahlstrom et al., 2015*, *2016*).

## Results

### Activation of full-length LapD results in quaternary structure changes

Previously, we showed that LapD has a reduced binding capacity for LapG in the absence of c-di-GMP (*Chatterjee et al., 2014*). This observation was based on an equilibrium-binding assay at a fixed LapD concentration and with LapG as the titrant. To further confirm a quantitative difference between c-di-GMP-bound and -unbound LapD with regard to LapG affinity, we developed a fluorescence anisotropy-based assay, which relies on LapG that is fluorescently labeled at the sole cysteine residue in the protein's active site (*Figure 1—figure supplement 2*). Titration of purified, detergent-solubilized LapD to a fixed concentration of fluorescent LapG yielded saturation-binding data, revealing an approximately 6-fold increase in LapG's apparent affinity for LapD when c-di-GMP is present. Interestingly, the LapD titrations used here reached comparable maximum binding with and without c-di-GMP (*Figure 1—figure supplement 2*), in contrast to our previous assays in which LapG was the titrant which showed distinct maxima for LapD–LapG binding in the presence or absence of c-di-GMP (*Chatterjee et al., 2014*). The difference in binding observed when the fixed and titrated proteins are swapped suggests that LapG displays a constant, saturable binding site, whereas LapD's binding capacity, and not only its affinity for LapG, is regulated by the dinucleotide.

To begin investigating the structural underpinnings of LapD switching in the context of the intact receptor, we initially employed size-exclusion chromatography coupled to multi-angle light

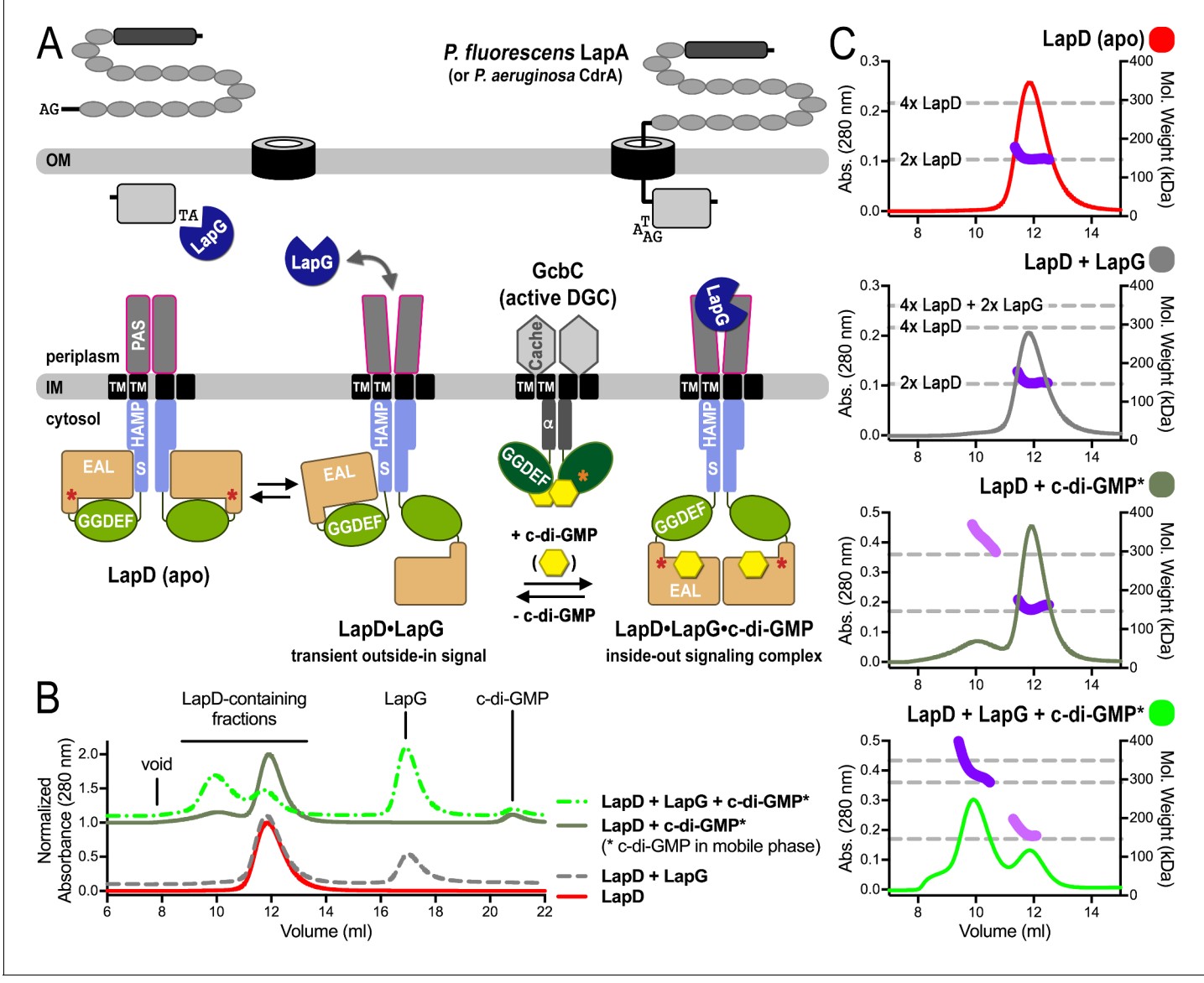

**Figure 1.** SEC-MALS reveals a switch of LapD dimers to dimer-of-dimers upon ligand binding. (**A**) Working model for c-di-GMP-dependent regulation of the periplasmic protease LapG via the inner membrane protein LapD. Concerted conformational changes expose a periplasmic binding site for LapG on LapD, sequestering the protease away from its substrates, the adhesin proteins LapA in *P. fluorescens* or CdrA in *P. aeruginosa*. (DGC, diguanylate cyclase; red/orange asterisks indicate interaction helices in LapD/GcbC). (**B**). Size-exclusion chromatograms for detergent-solubilized LapD in different states. Samples were prepared as described in the Material and Methods. (Asterisks: c-di-GMP was included in the mobile phase). (**C**) Molecular weight of LapD in solution. Peak fractions were analyzed by in-line SEC-MALS. (Absorbance at 280 nm: Traces colored according to (**B**); molecular weight determination: Dark and light purple dots; theoretical molecular weights based on sequence: Horizontal dashed lines.) Data are representative of two biological replicates using independent protein preparations.

The following figure supplements are available for figure 1:

**Figure supplement 1.** Mapping of physical interaction motifs onto the crystal structures of LapD and GcbC.

**Figure supplement 2.** C-di-GMP binding to LapD results in a 6-fold increase in apparent affinity for LapG.

**Figure supplement 3.** SEC-MALS analyses of protein–detergent complexes.

scattering (SEC-MALS). This approach enables assessment of the quaternary structure of detergent-solubilized, full-length LapD in response to c-di-GMP and LapG (*Figure 1B and C*, *Figure 1—figure supplement 3*). SEC-MALS is particularly advantageous for the study of membrane proteins because the respective scattering contributions of the protein and detergent components of the protein-detergent conjugate can be accurately deconvoluted to yield the molecular mass of the protein analyte (*Gimpl et al., 2016*) (*Figure 1—figure supplement 3C*). With this approach, purified apo-LapD was shown to be monodisperse and dimeric (*Figure 1C*, *Figure 1—figure supplement 3C*). Pre-incubation with LapG in the absence of c-di-GMP did not alter LapD's elution time nor oligomerization state (*Figure 1C*), indicating that the interactions between the c-di-GMP-unbound receptor and the protease observed in saturation-binding assays ((*Chatterjee et al., 2014*); *Figure 1—figure supplement 2*) are likely transient.

To probe how c-di-GMP influences LapD oligomerization, LapD was pre-incubated with dinucleotide and injected into the SEC-MALS system equilibrated with c-di-GMP in the mobile phase (*Figure 1B*). The LapD•c-di-GMP complex was predominantly dimeric, but a minor population of tetrameric LapD•c-di-GMP was also observed. Unexpectedly, when analyzed under identical conditions, the preformed LapG•LapD•c-di-GMP complex eluted predominantly as a higher-order species with a molecular weight corresponding to that of a LapD dimer-of-dimers (*Figure 1C*, *Figure 1—figure supplement 3C*). Omitting c-di-GMP in the mobile phase destabilized the complex, with LapD eluting as a dimeric species (*Figure 1—figure supplement 3A and C*). Together, these results provide compelling evidence that the LapG protease, together with c-di-GMP, plays a crucial role in the activation of LapD via induction of receptor oligomers.

## Cysteine crosslinking reveals multiple activation states of LapD

The analysis described above suggests that native LapD can adopt at least three global states: (1) dimeric apo-LapD, (2) dimeric LapD•c-di-GMP, and (3) a LapD•c-di-GMP•LapG dimer-of-dimers complex. To characterize these states further, we turned to cysteine crosslinking, a time-honored approach to assess the conformation and dynamics of proteins with high spatial resolution (*Bass et al., 2007*; *Butler and Falke, 1998*). In this approach, a covalent crosslink forms upon mild oxidation, catalyzed by copper phenanthroline (Cu(Phen)$_2$), only when two cysteine residues are in close spatial proximity (~2–3 Å). For visualization of crosslinked polypeptides in SDS-PAGE, we expressed LapD variants with a C-terminal monomeric superfolder-GFP (sfGFP) tag. Specific LapD–sfGFP detection in crude solubilized membrane fractions is based on in-gel fluorescence, with crosslinks introducing electrophoretic mobility shifts of the sfGFP-fusion protein. Furthermore, we mutated two native cysteine residues in the soluble domains of LapD (C$^{304}$ and C$^{397}$) to alanine and serine, respectively, to eliminate non-specific crosslinks. This LapD variant ('TMcys') retained the two native cysteine residues in the transmembrane helices (C$^{12}$ and C$^{158}$). Using a specific LapG photo-crosslinking assay (*Chatterjee et al., 2014*), we showed that LapD-TMcys was able to bind LapG in a c-di-GMP-dependent manner similar to wild-type LapD, as was a variant lacking all four native cysteine residues ('cysless') (*Figure 2—figure supplement 1A*).

Site-directed cysteine mutations were introduced into the TMcys–sfGFP fusion background in order to analyze LapD conformational changes in different states of activation. One central strategy in our studies is based on the mutation A$^{602}$C, which allows us to probe EAL domain dimerization and, in conjunction with S$^{229}$C, signaling (S) helix–EAL domain contacts, reporting on active and autoinhibited LapD, respectively (*Figure 2A*). We confirmed by SEC-MALS analysis that LapD–sfGFP, LapD$^{TMcys}$, and a representative variant with an engineered cysteine, LapD$^{TMcys}$-A$^{602}$C, formed constitutive dimers in their respective apo-states, similar to wild-type LapD, and that neither the sfGFP fusion nor cysteine mutagenesis perturbed the receptor notably (*Figure 2—figure supplement 1B*).

### Apo-LapD: only a fraction of apo-LapD protomers adopt the autoinhibited conformation at a given time

We first probed the apo-state of LapD in which the cytoplasmic EAL domain is thought to engage with the S helix (*Navarro et al., 2011*) (*Figure 2A*). In the corresponding crystal structure, residues S$^{229}$ of the S helix and A$^{602}$ of the EAL domain of the same polypeptide chain are in close proximity. Hence, cysteine mutations that are introduced at these positions allow us to assess their propensity to support disulfide formation as a reporter for the autoinhibited conformation of LapD. Upon

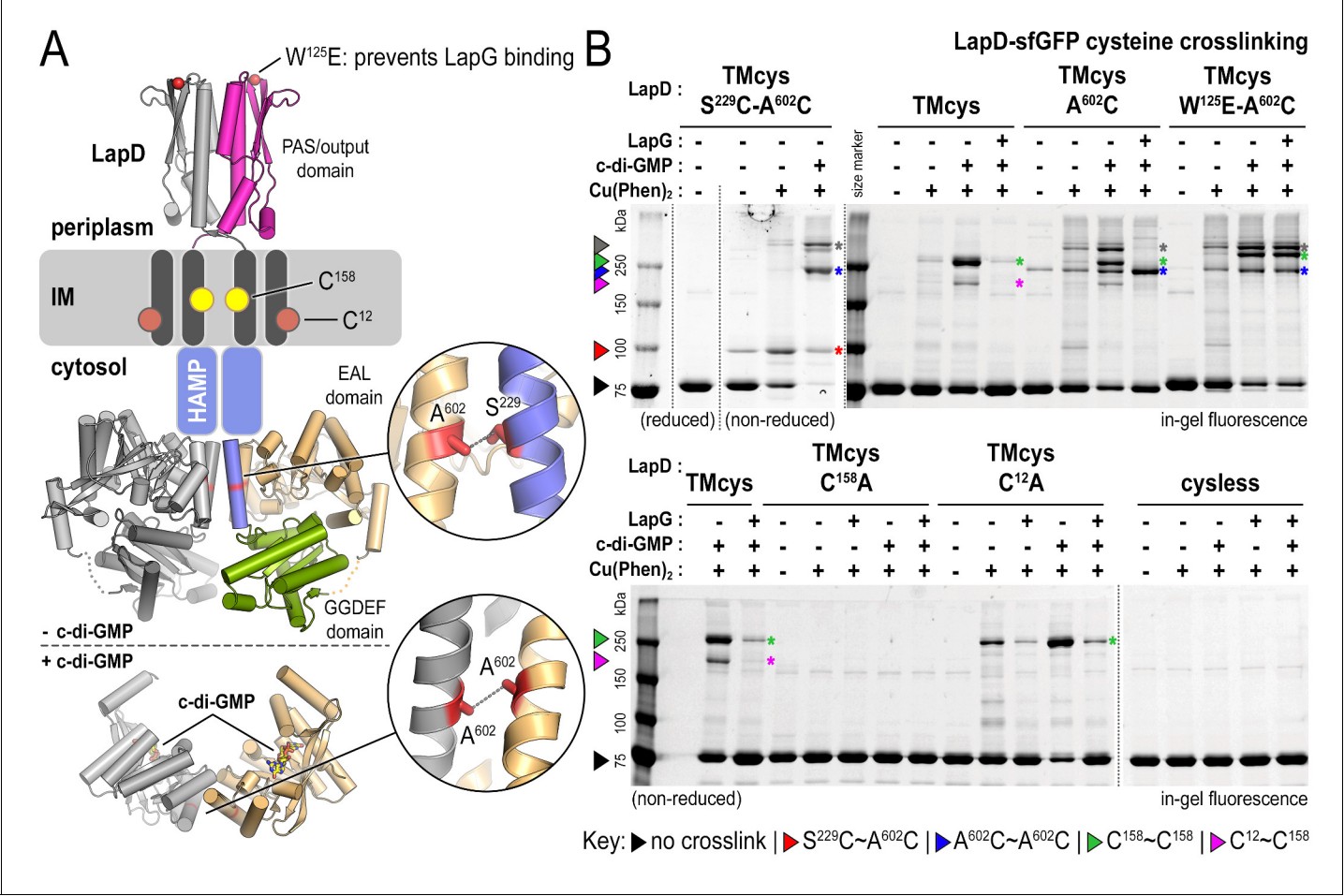

**Figure 2.** Cysteine crosslinking reveals distinct conformations of apo-, c-di-GMP, and c-di-GMP–LapG-bound LapD. (A) Overview of native and engineered cysteine residues used in crosslinking studies. The composite model of LapD shown is based on the available crystal structures of autoinhibited (top) and c-di-GMP-bound (bottom) domains. (B) Detection of intramolecular and intradimer disulfide bonds. Upper panel: LapD$^{TMcys}$–sfGFP variants with the indicated cysteine mutation. Colored arrows and asterisks mark specific crosslinking bands. Only the band with grey marking could not be assigned unambiguously to a specific crosslinking sites. Note: The W$^{125}$E variant of LapD migrates slightly slower than its native counterparts (also seen in *Figure 2—figure supplement 1A*). Lower panel: LapD$^{TMcys}$–sfGFP variants with native transmembrane cysteine residues mutated. Experiments with individual protein variants (wild-type and mutants) were repeated at least three times using independent protein preparations. Primary data reproduced here are representative of each replicate.

The following figure supplement is available for figure 2:

**Figure supplement 1.** Engineered LapD variants are functional and form constitutive dimers.

expressing LapD$^{TMcys}$-S$^{229}$C-A$^{602}$C (fused to sfGFP), there is a low, but detectable level of spontaneous disulfide-bond formation in the absence of oxidant, which was sensitive to reducing agent (*Figure 2B*; red arrow/asterisk). Short copper-catalyzed oxidation (~10 min) yields crosslinks of only about half of apo-LapD protomers. This result suggests that conformational heterogeneity exists within apo-LapD and only a certain population of protomers adopt the autoinhibited state at a given time.

## c-di-GMP-bound LapD: dinucleotide binding leads to EAL domain dimerization and is accompanied by reorientation of transmembrane helices

Addition of c-di-GMP to the S$^{229}$C-A$^{602}$C variant prior to copper oxidation abolishes the oxidant-dependent, intra-molecular crosslinking. Instead, new bands with slower electrophoretic mobility are

observed (*Figure 2B*; blue and grey markings), which account for the entire LapD population (minus the spontaneously crosslinked fraction that is locked in the autoinhibited state prior to oxidation). One of these bands (blue marking) is rationalized as a homotypic crosslink between corresponding cysteine residues at position 602, as residue $A^{602}$ is also central to the canonical EAL domain dimerization interface observed in the $LapD^{EAL}$•c-di-GMP complex structure (*Figure 2A*, bottom) (*Navarro et al., 2011*). This interpretation is supported by the presence of a crosslinking adduct with comparable electrophoretic mobility using $LapD^{TMcys}$-$A^{602}C$, which lacks the $S^{229}C$ mutation, and by the absence of such a band in $LapD^{TMcys}$, which lacks both $S^{229}C$ and $A^{602}C$, despite otherwise identical experimental conditions (*Figure 2B*).

We hypothesized that the origin of two other predominant bands with slower mobility in the $LapD^{TMcys}$-$S^{229}C$-$A^{602}C$•c-di-GMP sample stems from movements in the transmembrane helices upon activation, which may orient the native cysteine residues $C^{12}$ and/or $C^{158}$ into crosslinking-competent positions. Indeed, the exposure of c-di-GMP-activated $LapD^{TMcys}$ to oxidant gave rise to two crosslinking adducts with slower electrophoretic mobility (green and magenta markings, *Figure 2B*). This observation indicates that the transmembrane domain cysteine residues contribute to the disulfide-mediated banding pattern in the presence of c-di-GMP. To identify explicitly the relevant crosslinks involving the transmembrane helices, we mutated these two native cysteine residues individually to alanine and subjected them to oxidative crosslinking. No crosslinked bands were observed in the case of $LapD^{TMcys}$-$C^{158}A$, whereas $LapD^{TMcys}$-$C^{12}A$ supported a single crosslinking adduct (*Figure 2B*, bottom panel). Hence, the dominant adduct (green marking) pertains to covalent bonds between corresponding $C^{158}$ residues in a LapD dimer, whereas the lower, weaker band (magenta marking) stems from a crosslink between $C^{12}$ and $C^{158}$ (*Figure 2*, bottom panel).

Taken together, the shift to a LapD species with slower electrophoretic mobility upon incubation with c-di-GMP can, for the most part, be explained by cysteine crosslinking caused by movements in the transmembrane helices (green/magenta markings) and EAL domain dimerization (blue marking), as well as a subpopulation presumably containing multiple crosslinks (grey marking) (*Figure 2B*).

## LapD•c-di-GMP•LapG: LapG contributes to LapD activation by inducing distinct conformational changes

Using $A^{602}C$ crosslinking as an analytical tool, we next probed the effect of LapG on the activation of LapD complexes. Notably, crosslinking via corresponding $A^{602}C$ residues in the LapD•c-di-GMP•LapG complex increases relative to the level observed in the absence of LapG (*Figure 2B*, top panel; blue markings). This apparent close proximity of $A^{602}C$ residues is indicative of the formation of canonical, c-di-GMP-dependent EAL domain dimers (*Navarro et al., 2011*; *Barends et al., 2009*; *Minasov et al., 2009*; *Sundriyal et al., 2014*). Concomitantly, LapG addition prevented $C^{158}$–$C^{158}$ (green marking) and $C^{12}$–$C^{158}$ (magenta marking) crosslinks between transmembrane helices, as well as other, higher molecular weight (gray marking) crosslinks (*Figure 2B*). Addition of LapG in the absence of c-di-GMP also causes a conformational change that is not compatible with $C^{158}$–$C^{158}$ crosslinking (*Figure 2B*, bottom panel), suggesting that LapG contributes to LapD structural transitions independently of c-di-GMP. Mutating the main anchor residue for LapG on LapD's periplasmic domain binding, $W^{125}$ (*Chatterjee et al., 2014*; *Navarro et al., 2011*; *Chatterjee et al., 2012*), abolished these LapG-induced effects on cysteine crosslinking but not those dependent on c-di-GMP (compare $LapD^{TMcys}$-$A^{602}C$ and $LapD^{TMcys}$-$W^{125}E$-$A^{602}C$; *Figure 2B*, top panel). This control establishes specificity and implicates the crystallographic LapD–LapG interface in LapG's effect on LapD conformation.

Together with the SEC-MALS data, these observations unequivocally demonstrate that the transition from dimeric LapD•c-di-GMP to tetrameric LapD•c-di-GMP•LapG is coincident with EAL domain dimerization. Interestingly, only ~50% of the EAL domains appear to be engaged in dimers, suggesting that like LapD in the autoinhibited state, activated LapD is based on asymmetric LapD dimers as the minimal unit.

## EAL domain dimerization mediates the formation of LapD dimer-of-dimers

Our previous model assumed that EAL domain dimerization occurs within a LapD dimer (*Figures 1A* and *3A*, left panel), a notion that needed to be revisited in light of the LapD dimer-of-dimers observed in the presence of LapG and c-di-GMP (*Figure 1C*). An alternative model consistent with

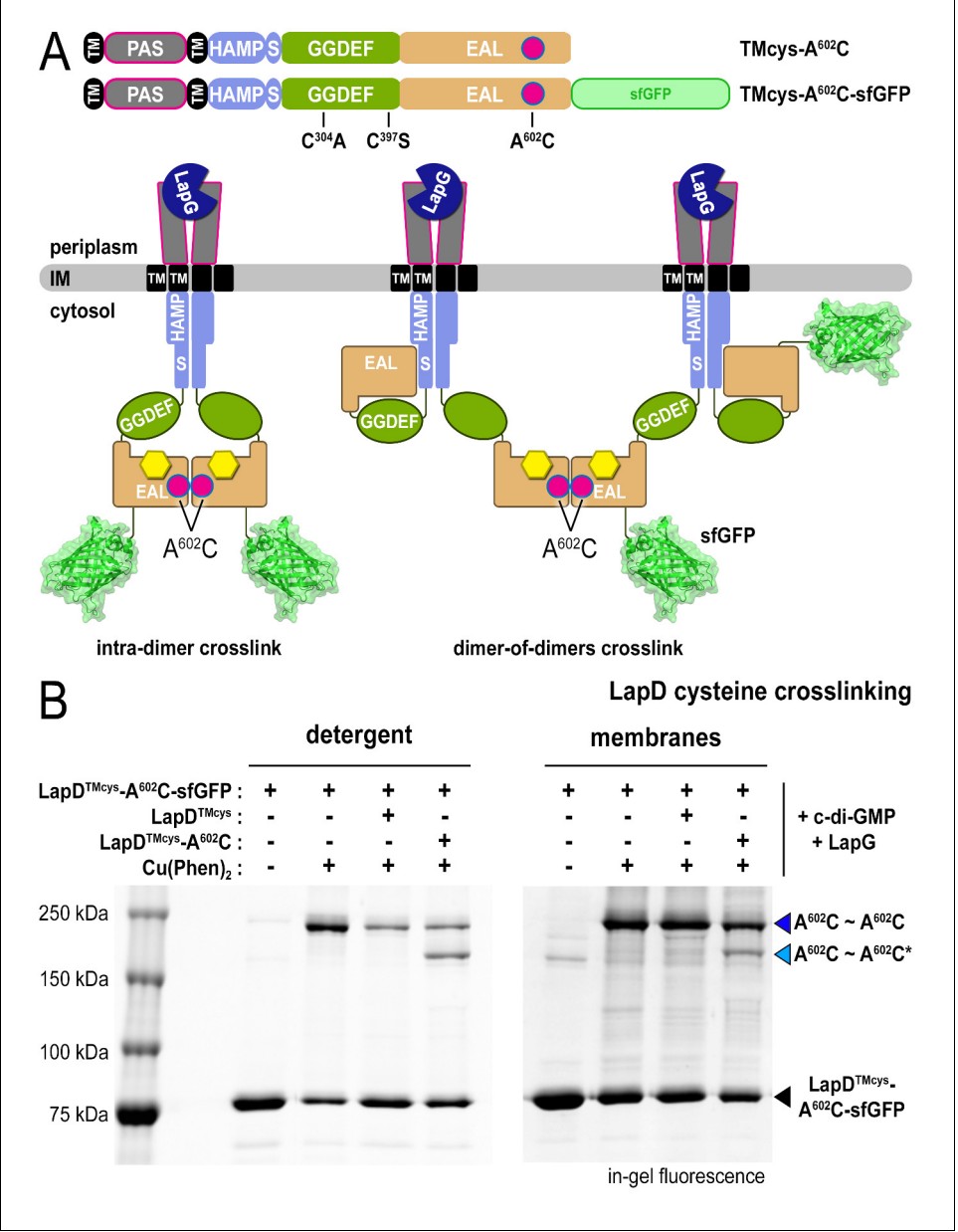

**Figure 3.** LapD•c-di-GMP•LapG dimer-of-dimers also form in a membrane environment. (**A**) Two LapD constructs, one genetically fused to sfGFP and one non-fluorescent, both harboring the A$^{602}$C mutation, were engineered and expressed separately. In previously proposed models, EAL domain dimerization was thought to occur within a single LapD dimer (left). Under this model, one would only expect a single band shift corresponding to c-di-GMP- and LapG-activated LapD–sfGFP dimers upon oxidation, even if the sfGFP-fused and non-fluorescent variants were present in the membrane. Alternatively, the EAL domains could dimerize across two LapD dimers (right) to form a dimer-of-dimers. Under the latter model, when the two constructs are mixed, activated with c-di-GMP and LapG, oxidized with a disulfide-promoting copper catalyst, denatured in SDS-PAGE, and imaged by in-gel fluorescence, a faster migrating covalent heterodimeric adduct consisting of LapD–sfGFP and LapD (dark) should be observed in addition to the slower migrating complex containing just LapD–sfGFP homodimeric adduct. (**B**) Complex formation is mediated by dimerization of EAL domains across two LapD dimers rather than within the same LapD dimer. The two LapD variants shown in (**A**) were expressed separately. Crosslinking via corresponding A$^{602}$C residues was induced after incubation with c-di-GMP and LapG, either in detergent-solubilized samples (left panel) or upon fusing membrane fractions from the two cultures (right panel). In both detergent and membranes, SDS-PAGE analysis of this crosslinking experiment shows that a heterodimeric adduct containing both LapD– sfGFP and non-fluorescent LapD (lighter blue triangle; asterisk denotes residue from a non-fluorescent LapD) is

*Figure 3 continued on next page*

*Figure 3 continued*

observed in addition to a species containing only LapD–sfGFP (darker blue triangle). In detergent, non-fluorescent LapD lacking the A$^{602}$C mutation serves as a competitor, reducing LapD-A$^{602}$C–sfGFP crosslinking efficiency. Representative data from two independent, biological replicates are shown.

the above and previous findings (*Navarro et al., 2011*) is one in which EAL domains bridge two LapD dimers rather than interacting within the same LapD dimer (*Figure 3A*, right panel). To distinguish between these two possibilities, we mixed LapD variants with distinct molecular weight, LapD–sfGFP and non-fluorescent LapD, which separately form constitutive dimers in their respective apo-state (*Figure 2—figure supplement 1B*). As seen before (e.g. *Figure 2B*), oxidation of LapD$^{TMcys}$-A$^{602}$C–sfGFP in the presence of c-di-GMP and LapG produced a crosslinking band that migrates just below the 250 kDa marker (*Figure 3B*; dark blue arrow). Addition of LapD$^{TMcys}$ lacking sfGFP prior to oxidation decreases crosslinking efficiency, indicating a competitive effect between LapD$^{TMcys}$-A$^{602}$C–sfGFP and LapD$^{TMcys}$ dimers. To unambiguously demonstrate that this competition is due to the formation of a LapD dimer-of-dimers, we replaced non-fluorescent LapD$^{TMcys}$ with non-fluorescent LapD$^{TMcys}$-A$^{602}$C. Upon oxidation, we observe two crosslinking bands that correspond to LapD$^{TMcys}$-A$^{602}$C–sfGFP and LapD$^{TMcys}$-A$^{602}$C–sfGFP•LapD$^{TMcys}$-A$^{602}$C dimers-of-dimers (dark and light blue arrow, respectively), for which the difference in molecular weight after SDS denaturation is roughly a single sfGFP moiety. Importantly, these specific crosslinking events occur readily with detergent-solubilized LapD at low protein concentration (~1 μM). Qualitatively similar results were obtained when proteins are embedded in a lipid bilayer of fused membrane fractions from cells separately expressing LapD$^{TMcys}$-A$^{602}$C–sfGFP or non-fluorescent LapD$^{TMcys}$ variants (see Material and Methods; *Figure 3B*). In summary, these data provide a structural and mechanistic basis for the dimer-of-dimers observed in the SEC-MALS data.

## Modeling of small angle X-ray scattering (SAXS) data elucidates the conformational transitions upon LapD activation

In order to obtain further structural evidence for LapD's global conformational changes, we modeled SAXS data collected on detergent-solubilized LapD in four distinct states (*Figure 4A*; *Figure 4—figure supplement 1*): (1) apo-LapD trapped in the c-di-GMP-insensitive, fully autoinhibited conformation via an intramolecular crosslink between S$^{229}$C and A$^{602}$C (see Materials and Methods, *Figure 2*, *Figure 2—figure supplement 1A*, and *Figure 4*); (2) Apo-LapD; (3) LapD•c-di-GMP; (4) LapD•c-di-GMP•LapG. The first two states were stably monodisperse after purification, and could be analyzed directly. The two c-di-GMP-containing states, however, contained minor peaks based on our SEC-MALS data (*Figure 1C*). To avoid complications associated with such polydisperse samples and to capitalize on the observation that the predominant species could be separated by gel filtration, these samples were injected into an in-line size-exclusion chromatography system mirroring our SEC-MALS experiments. Guinier analysis of these samples collected on the SEC-SAXS setup (*Malaby et al., 2015*) revealed that they were monodisperse, and were therefore suitable for further analysis. Interpretation of the SAXS data was facilitated by having experimentally determined crystallographic structures of isolated domains of LapD and by the recently developed program MEMPROT, which models a detergent corona around transmembrane domains (*Pérez and Koutsioubas, 2015*).

Analysis of distance distribution functions suggests a slightly more compact structure of the trapped-inactive LapD with a measurably shorter $D_{max}$ and smaller Porod volume compared to native apo-LapD (*Figure 4B*, left panel; *Figure 4—source data 1*). These observations are reflected in *ab initio* three-dimensional envelopes in which the trapped-inactive LapD lacks the extended density toward the bottom of the molecule seen in native apo-LapD (*Figure 4C*, dotted circles). Manually docking individual LapD domains suggests that the trapped-inactive LapD envelope has dimensions consistent with both protomers' adopting the autoinhibited conformation that was observed in the crystal structure of this module. On the other hand, the native apo-LapD envelope is consistent with a heterogeneous state in which one protomer adopts the autoinhibited conformation while the other half-site of the dimer can be modeled as a mixture of autoinhibited and extended conformations. In this proposed extended conformation, the EAL domain dislodges from the S helix

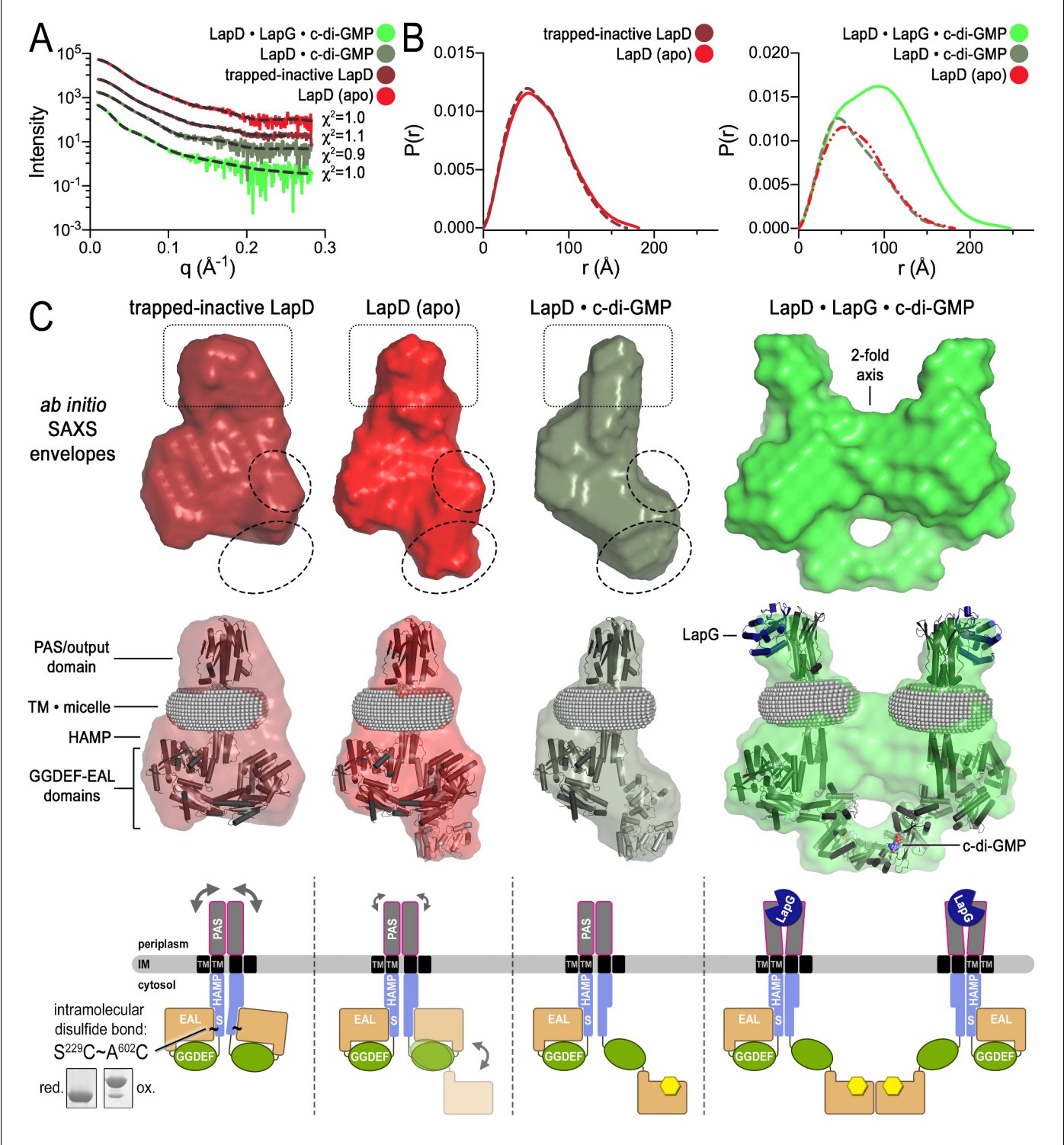

**Figure 4.** Modeling of SAXS data for distinct LapD states illustrates the conformational changes upon receptor activation. (**A**) Primary SAXS data. Solid lines, experimental scattering curves of LapD in the states indicated; dashed lines, theoretical scattering curves of the three-dimensional envelopes shown in panel (C) with $\chi^2$ values listed to the right. (**B**) Real-space pair-wise distance distribution functions for each state of LapD. (**C**) Modeling of SAXS data. Top: Ab initio three-dimensional envelopes calculated on the basis of the experimental scattering data. Dotted circles and boxes highlight areas of density that change between different states. Middle: Crystal structures of individual domains of LapD docked manually into the envelopes depict interpretations of the *ab initio* envelope models. Gray spheres represent the detergent corona that surrounds the transmembrane domain. *Figure 4 continued on next page*

*Figure 4 continued*

Bottom: Cartoon models of LapD domain movements in each state based on the SAXS data (bottom left inset: SDS-PAGE of purified trapped-inactive LapD used for SAXS analysis). Source files of SAXS data and envelope data are available in *Figure 4—source data 2*.

The following source data and figure supplements are available for figure 4:

**Source data 1.** Statistics associated with the analysis of LapD small-angle X-ray scattering data.

**Source data 2.** SAXS data and envelope model files related to *Figure 4*.

**Figure supplement 1.** Fitting of various models to LapD small-angle X-ray scattering data.

**Figure supplement 2.** Simulated annealing and rigid-body fitting of crystal structures of individual LapD domains to full-length LapD small-angle X-ray scattering data.

**Figure supplement 3.** Ensemble optimization method (EOM) analysis of the LapD SAXS data.

**Figure supplement 4.** Proposed mechanism for activation of the LapD receptor.

to expose the c-di-GMP binding site to solvent. This notion of a partially dislodged EAL domain in one protomer of native apo-LapD is consistent with the S helix–EAL domain cysteine crosslinking data discussed above, in which only ~50% of the protomers are engaged upon short exposure to the disulfide-promoting copper catalyst.

In accordance with this analysis, addition of c-di-GMP has a marginal effect on native LapD's $D_{max}$, indicating that part of apo-LapD already adopts an extended conformation (*Figure 4B*). However, the shape of the distance distribution function confirms c-di-GMP-dependent conformational changes relative to apo-LapD (*Figure 4B*, right panel). The modeled envelope of c-di-GMP-bound LapD reveals a more pronounced density for a disengaged EAL domain and a lessening of features that would correspond to the EAL domain in its S helix-engaged, autoinhibited conformation (*Figure 4C*). Notably, these global changes appear to be restricted to one half-site of a LapD dimer, consistent with our crosslinking data (*Figure 2*) and the lack of apparent symmetry in the SAXS data for the constitutive dimeric states discussed so far. Even the trapped-inactive LapD dimer could not be modeled with two-fold symmetry, indicating that crosslinking stabilizes or introduces some sort of a strained conformation (*Figure 4C*). Alongside the trend of conformational changes in the cytosolic module of LapD from trapped-inactive to c-di-GMP-bound dimer, we noticed volume effects in the periplasmic domain (boxed area, *Figure 4C*), which we interpret as decreasing dynamics of this domain from the trapped-inactive, to the apo- and finally the c-di-GMP-bound state.

Lastly, the distance distribution function and envelopes generated from LapD•c-di-GMP•LapG SAXS profiles yielded dimensions and overall shapes consistent with two LapD•c-di-GMP dimers engaged via one of their EAL domains (and bound to LapG) (*Figure 4B and C*). At the same time, this complex appears to be less well defined as the normalized spatial discrepancy (NSD) values, an assessment of the similarity of the individual models used to calculate the average envelope, are higher than those of the other three states (*Figure 4—source data 1*).

Another way to model SAXS data relies on simulated annealing-rigid-body fitting of individual domains on the basis of high-resolution structures (*Petoukhov and Svergun, 2005*). This analysis yielded models that are compatible with the experimental data, and these models agreed well with the above interpretations based on envelope calculations for the trapped-inactive, the LapD•c-di-GMP complex, and the fully active LapD•c-di-GMP•LapG complex (*Figure 4—figure supplement 2*). Interestingly, the native apo-LapD was modeled as an intermediate between a half-open and a fully inhibited conformation, depicting a single average conformation.

The above structure-based, rigid-body modeling creates a single solution, even for more dynamic or heterogeneous states such as native apo-LapD. By contrast, the Ensemble Optimization Method (EOM) enables selection of an ensemble of conformations from a pool of random conformers based on fitting theoretical and experimental scattering profiles (*Bernadó et al., 2007*; *Tria et al., 2015*). For this approach, we generated two libraries of 10,000 structures each, one in which the linker

connecting the GGDEF and EAL domains of one protomer of LapD was allowed to be flexible, and a second in which this linker was flexible for both protomers. Results from this EOM analysis are analogous to those discussed above in that the native apo-LapD adopts an extended conformation (based on the $R_g$ and $D_{max}$ of the selected ensemble models) more frequently than the trapped-inactive form. The LapD•c-di-GMP state is enriched in conformations that are more extended than both the trapped-inactive and the native apo-form (*Figure 4—figure supplement 3*).

In summary, results from the three independent approaches reporting on conformational changes upon LapD activation or inactivation – SEC-MALS, site-specific crosslinking, and SAXS-based modeling – are consistent with each other and allow us to distinguish four distinct states of the receptor (*Figure 4C*, bottom panel). The structural transition from a low-to-high affinity state of LapD for the protease LapG appear to be driven by independent inputs via c-di-GMP binding to the cytosolic module and LapG binding to LapD's periplasmic domain.

## Discussion

Cellular signaling systems must transmit information with high specificity and do so reversibly, avoiding high-affinity or excessively stable complexes that would result in unregulated signaling events. In the present study, we uncovered structural events that ensure exquisite control of a bacterial biofilm signaling switch, the conserved second messenger receptor LapD, based on the characterization of the intact membrane protein. LapD sequesters and thus regulates its cognate periplasmic protease LapG in response to rising c-di-GMP levels in the cytosol, a mechanism that has been described as inside-out signaling (*Newell et al., 2009*, *2011b*). Yet, new evidence presented here indicates that both ligands, c-di-GMP and LapG on either side of the inner bacterial membrane, contribute to the conformational switch that establishes a signaling competent state. The distinct response as a result of two independent inputs is a hallmark of coincidence detection, achieved here through bidirectional signaling across the membrane.

The mechanistic implications of these new findings for LapD regulation are manifold. We show that a LapD dimer is already inherently asymmetric even in the absence of any ligands. One half-site of an apo-LapD dimer adopts a conformation akin to the crystal structure of the autoinhibited cytosolic module, characterized by an occluded c-di-GMP binding site (*Navarro et al., 2011*). The other half-site appears to sample an open conformation, in which the EAL domain is disengaged from LapD's regulatory S helix. Such a conformation would prime LapD for c-di-GMP binding, which, in turn, would shift the equilibrium to the more extended state, thus preventing autoinhibition. The reason why one protomer appears more dynamic than the other is not well understood, but could indicate anti-cooperative effects in oligomeric receptors analogous to mechanisms described for other bacterial receptors (*Maksay and Tőke, 2014*). In support of this, we previously showed that only one equivalent of c-di-GMP is required per LapD dimer for maximal LapG binding, and that the high-affinity LapG-binding state of the periplasmic domain of LapD is asymmetric (*Chatterjee et al., 2014*). Both of these features could manifest from the proposed anti-cooperative nature of activation.

In several other receptor systems, in particular those utilizing HAMP domains as an integral relay module, one model for signal transmission is based on an alternating static-to-dynamic transition in which tighter packing in one protein element drives other elements into a more loosely packed, dynamic state (known as the 'yin-yang' model) (*Swain et al., 2009*; *Zhou et al., 2009*; *Parkinson, 2010*). Such a model of coupled dynamics is enticing for LapD, in which compact autoinhibited cytoplasmic domains give rise to a more dynamic output domain capable of only transiently binding LapG (*Figures 2B* and *4C*). By contrast, activation of the complex with c-di-GMP results in a disengaged EAL domain and in more compact output domains that are competent to bind LapG more tightly. This notion is supported by our modeling of the SAXS data, which indicates that the output domain becomes more well-defined upon LapD activation by c-di-GMP.

At the same time, we also demonstrate that c-di-GMP is only a partial activator and that c-di-GMP-bound LapD differs structurally from high-affinity complexes comprising the dinucleotide, receptor, and protease. This observation indicates a distinct signaling input into LapD via LapG. The simultaneous binding of c-di-GMP and LapG to LapD's cytosolic and periplasmic domains, respectively, establishes a conformation in which the EAL domains dimerize through a canonical interface (*Navarro et al., 2011*; *Barends et al., 2009*; *Minasov et al., 2009*; *Sundriyal et al., 2014*). In

contrast to previous models, this interaction does not occur within a dimer, but across two LapD dimers. This change in quaternary structure, the formation of a LapD dimer-of-dimers, is apparent in all three, independent approaches reported on here. An exact molecular mechanism explaining this long-range allosteric regulation of LapD resulting in a higher oligomeric, high-affinity complex remains enigmatic, but it appears that the HAMP domain functions as an integrator of bilateral signaling across the membrane. In this model, the HAMP domain would prevent full LapD activation if the receptor were to engage with only one of the ligands, LapG or c-di-GMP, on either side of the membrane.

The periplasmic input — LapG binding to LapD's output domain — is likely a vestigial feature derived from evolutionarily related, active enzymes that have domain architecture similar to that of LapD, which produce or degrade cellular c-di-GMP upon sensing environmental stimuli through an outside-in signaling mechanism. Some parallels can be drawn to the YfiBNR system in *P. aeruginosa* (*Malone et al., 2012*, *2010*). YfiN is a transmembrane, GGDEF-domain-containing diguanylate cyclase with a juxtamembrane HAMP domain and a periplasmic PAS domain. The cyclase-inactive state is stabilized through interactions of YfiN's periplasmic domain with YfiR, a periplasmic protein. An environmental signaling cue, possibly cell wall or redox stress, leads to the dissociation of YfiR, which in turn increases YfiN's cyclase activity and — as a result — cellular c-di-GMP levels. While YfiN regulation probably relies on its integral HAMP domain and periplasmic ligand binding (*Giardina et al., 2013*), direct juxtaposition with LapD's mechanism is hindered by a gap in our understanding regarding the structure of the full-length diguanylate cyclase.

Annotation of close to 2,000 proteins containing a triple HAMP-GGDEF-EAL domain unit akin to LapD, many of which are predicted to be active diguanylate cyclase or phosphodiesterases, highlights the prevalence of this module in c-di-GMP signaling networks. It is possible that the underlying molecular mechanisms for the regulation of these proteins follows the paradigm that we established here for LapD. Indeed, asymmetry, like that we described for the LapD output domain-LapG complex (*Chatterjee et al., 2014*), is common among outside-in signaling membrane proteins, including HAMP domain-containing chemotaxis receptors and histidine kinases (*Maksay and Tőke, 2014*; *Milburn et al., 1991*; *Neiditch et al., 2006*; *Yeh et al., 1996*). For example, binding of the quorum-sensing signal AI-2 to the periplasmic domain of the histidine kinase complex LuxPQ induces asymmetry in its periplasmic domains (*Neiditch et al., 2006*). Likewise, ligand-induced asymmetry in Tsr/Tar chemosensory receptors promotes the oligomerization of three identical, but functionally non-equivalent, cytoplasmic core domains (*Briegel et al., 2012*; *Li et al., 2013*; *Liu et al., 2012*). It will be interesting to investigate the conservation of the molecular mechanisms established for LapD in other c-di-GMP-signaling proteins. Such comparisons are currently hampered by limited structural insight for other HAMP-GGDEF-EAL domain-containing proteins and a lack of knowledge regarding the ligands that control their activity and cellular function.

One of the major open questions in the field pertains to mechanisms that explain signaling specificity, that is, the non-equivalent physiological outcome of diguanylate cyclases or phosphodiesterases despite identical catalytic activity (*Dahlstrom et al., 2015*, *2016*; *Ha et al., 2014*; *Kulasakara et al., 2006*; *Newell et al., 2011a*). Consistently, protein–protein interactions between enzymes that make or break c-di-GMP and c-di-GMP-specific receptors have been implicated (*Abel et al., 2011*; *Lindenberg et al., 2013*; *Guzzo et al., 2013*; *Ryan and Dow, 2010*; *Ryan et al., 2012*), including the GcbC diguanylate cyclase and the LapD receptor pair (*Dahlstrom et al., 2015*, *2016*). Nevertheless, it was not entirely clear how these two membrane proteins might interact, as the motifs identified as mediating the physical contacts are either buried or sterically not accessible in a simple LapD dimer model (*Figure 1A*, *Figure 1—figure supplement 1*). Our revised model, based on the observation of asymmetric LapD dimers in the apo- and c-di-GMP-bound states, as well as the symmetric LapD dimer-of-dimers, reconciles this conundrum. These dimers have in common an open configuration that could provide a signaling platform for heterologous interactions with other proteins such as GcbC (*Figure 4—figure supplement 4*). Since apo-LapD samples the extended state, this model is also consistent with the observation that GcbC interacts with both apo- and constitutively active variants of LapD (*Dahlstrom et al., 2015*). The structural impact of the GcbC–LapD interaction on the c-di-GMP receptor remains to be assessed, but it is conceivable that GcbC could stabilize a LapD dimer-of-dimers and/or induce a conformation similar to that of the LapG–c-di-GMP-bound receptor. Regardless, the fact that LapD can adopt these many different conformations allows it to present unique interfaces for direct interaction with

other proteins for proper regulation. Such a mechanism for imparting strict specificity to an otherwise ubiquitous second messenger molecule has significant implications. The concepts of an open signaling platform, asymmetry as a vital feature for regulation, and coincidence detection could also be relevant for diguanylate cyclases and phosphodiesterases that require exquisite control and trigger specific signals in the background of a larger c-di-GMP signaling network. Conceivably, interventional strategies could be devised to target specific c-di-GMP cyclases, phosphodiesterases or receptors, despite these proteins all having similar folds or catalytic activity. Hence, revealing the language of these signaling networks will be crucial for controlling and treating chronic infections caused by biofilm-forming pathogens.

# Materials and methods

## Cloning, expression and purification of *Pseudomonas fluorescens* Pf0-1 LapD and LapG

Expression, membrane isolation and purification of non-fluorescent LapD and LapD fused C-terminally with monomeric superfolder green fluorescent protein (sfGFP containing the $V^{206}K$ mutation that prevents sfGFP dimerization) were performed as previously described (*Chatterjee et al., 2014*). LapG was expressed and purified as previously described (*Boyd et al., 2012*) with the exception that LapG was gel-filtered in the same buffer as LapD. LapG containing the UV photoactivatable crosslinking non-natural amino acid *para*-azidophenylalanine at position $V^{205}$ was expressed and purified as previously described (*Chatterjee et al., 2014*).

## Fluorescence anisotropy binding assay

The single native cysteine in LapG ($C^{135}$) was labeled with Oregon Green 488 maleimide (Molecular Probes) at a 1.5-fold molar excess of dye over protein (150 μM and 100 μM, respectively) for five minutes. Excess dye was removed via a NAP-5 desalting column (GE). Labeling efficiencies ranged between 45% and 55%. Anisotropy from a reaction consisting of 1 μM labeled LapG and LapD (varied between 1–36 μM) without or with c-di-GMP (100 μM) was measured with a Fluoromax-4 spectrofluorometer (Horiba Scientific, Edison, NJ) using 496 nm excitation and 515 nm emission wavelengths, 5 nm slit widths, 0.5 s integration time, and L-Type polarizers.

## Size-exclusion chromatography coupled to multi-angled light scattering analysis (SEC-MALS)

For apo-LapD, freshly purified protein was concentrated to ~10 mg/ml (~140 μM) and 40 μl was subjected to SEC using a Superdex 200 10/300 column (GE Life Sciences) equilibrated in SEC-MALS buffer (25 mM Tris-HCl pH 7.6, 250 mM NaCl, 0.005% (w/v) LMNG and 0.04% (w/v) CHAPS). For the LapD•c-di-GMP complex, protein at ~2 mg/ml (~28 μM) was incubated with 100 μM c-di-GMP on ice for 20 min and then concentrated ~5-fold prior to injection onto the same SEC column pre-equilibrated with MALS buffer containing 20 μM c-di-GMP. For the LapD•c-di-GMP•LapG complex, samples were prepared identically, except that a 3-fold molar excess of LapG over LapD dimer was added prior to concentration. The SEC was coupled to a static 18-angle light scattering detector (DAWN HELEOS-II), a UV detector, and a refractive index detector (Optilab T-rEX) (Wyatt Technology, Santa Barabara, CA). Data were collected every second at a flow rate of 0.5 ml/min. Data analysis of the protein–detergent complex was carried out using the protein-conjugate analysis package as part of the program ASTRA, yielding the reported molar masses of the protein components. For these calculations, a detergent dn/dc of 0.172 was used, which was determined experimentally for a CHAPS/LMNG (8:1) micelle following the method described by Strop and Brunger (*Strop and Brunger, 2005*) (*Figure 1—figure supplement 3B*). Extinction coefficients of 67,000 $M^{-1} \cdot cm^{-1}$, 50,100 $M^{-1} \cdot cm^{-1}$, and 26,100 $M^{-1} \cdot cm^{-1}$ for LapD, LapG and c-di-GMP, respectively, were used. Monomeric BSA (Sigma, St. Louis, MO) was used for normalization of the light-scattering detectors and for data quality control.

## Cysteine crosslinking

Native LapD contains four cysteines: two in the transmembrane domain ($C^{12}$ and $C^{158}$) and two in the soluble cytoplasmic domains ($C^{304}$ and $C^{397}$). Unless otherwise noted in the text, all variants of

LapD used for cysteine crosslinking retained the two native transmembrane cysteines while the two cytoplasmic cysteines were mutated to alanine and serine, respectively. These mutations did not alter the ability for LapD to bind LapG in a c-di-GMP-dependent manner (*Figure 2—figure supplement 1A*) compared to native protein. The mutations or sfGFP fusion had no significant effect on the SEC profiles of the proteins or the oligomeric state of the apo-proteins as shown by SEC-MALS analysis (*Figure 2—figure supplement 1B*). The positions at which non-native cysteine point mutations were introduced via site-directed mutagenesis are as described in the text.

To aid in visualizing crosslinked protein, LapD-sfGFP fusion proteins were used. This allows us to perform experiments in a non-purified system and to visualize LapD crosslinking patterns by in-gel fluorescence after SDS-PAGE (*Chatterjee et al., 2014*). Briefly, to prepare protein for cysteine crosslinking experiments, *Escherichia coli* cell membranes containing expressed variants of LapD–sfGFP were prepared as described previously (*Chatterjee et al., 2014*) and solubilized in 25 mM Tris-Cl pH 7.6, 150 mM NaCl, 5% glycerol, 2 mM $MgCl_2$, 2 mM $CaCl_2$, 1% (w/v) LMNG and 0.2% (w/v) CHAPS for 90 min at 4°C. Insoluble material was pelleted by ultracentrifugation at 150,000 x g for 20 min. The supernatant containing the detergent-solubilized protein was collected and concentrations of solubilized LapD–sfGFP were determined by measuring the absorbance at 488 nm using an extinction coefficient of 83,000 $M^{-1}$ $cm^{-1}$. To prepare the disulfide-promoting catalyst, 200 mM 1,10-phenanthroline was dissolved in ethanol and 200 mM $CuSO_4$ was prepared in water. Immediately before use, the stocks were added together in water in the appropriate ratios to yield 5 mM $Cu(Phen)_2$ catalyst (i.e. 5 mM $CuSO_4$ and 10 mM phenanthroline). For each reaction, 1 μM LapD dimer was pre-incubated with (or without) 3 μM LapG and 20 μM c-di-GMP for 15 min at room temperature, at which point copper catalyst was added to 100 μM and incubated for an additional 10 min. Samples were quenched by the addition of 10 mM EDTA and 20 mM N-ethylmaleimide, and immediately analyzed by SDS-PAGE on an 8% (w/v) acrylamide gel. Gels were imaged by fluorescence on a Bio-rad ChemiDoc system.

To assess LapD crosslinking at the EAL domain interface in fused membranes, membranes from *E. coli* cells expressing LapD-sfGFP and non-fluorescent LapD were prepared individually, resuspended in 25 mM Tris-Cl pH 7.6, 150 mM NaCl and 5% glycerol, and then mixed in equal volumes. LapG and c-di-GMP were added to final concentrations of 50 μM and 100 μM, respectively. A sub-solubilizing concentration of TritonX-100 (0.05% (v/v)) was added to the samples (*Dezi et al., 2013*; *Kragh-Hansen et al., 1998*; *Paternostre et al., 1988*; *Urbaneja et al., 1988*) and the membranes were fused by brief sonication. The insoluble material was pelleted by ultracentrifugation at 150,000 x g for 20 min. These fused, pelleted membranes were resuspended in the same buffer lacking detergent, and washed three times to remove traces of detergent prior to cysteine crosslinking, which proceeded as described above.

## Small angle X-ray scattering (SAXS) analysis

Preparation of LapD (without sfGFP) for SAXS analysis was performed exactly as previously described (*Chatterjee et al., 2014*) except that the final gel-filtration was conducted in SEC-MALS buffer. To generate the inactive, fully autoinhibited form of LapD (TMcys $S^{229}C/A^{602}C$), we noticed that in a purified system the amount of crosslinking between the S helix and EAL domain could be increased to >90% using exposures to the copper catalyst that were notably longer than those described above, presumably because this gives the receptor more time to sample the autoinhibited state and to become covalently trapped as such (*Figure 4C*, inset). Indeed, this covalently locked, fully autoinhibited form of LapD is functionally unresponsive to c-di-GMP (*Figure 2—figure supplement 1A*). This variant was prepared in an identical manner as native LapD, except that immediately after desalting into low-salt buffer and prior to gel-filtration, 400 μM copper phenanthroline was added and incubated for 75 min at room temp. The reaction was quenched with 10 mM EDTA, concentrated, and injected onto the Superdex 200 gel filtration column equilibrated with SEC-MALS buffer.

All protein samples were prepared fresh immediately before data collection and centrifuged at 150,000 x g at 4°C for 20 min prior to analysis. Buffer that flowed through the final size-exclusion column prior to protein elution was used for buffer subtraction. SAXS experiments were carried out at the Cornell High Energy Synchrotron Source (CHESS, beamline G1) at an electron energy of 9.9 KeV (*Acerbo et al., 2015*; *Skou et al., 2014*). For native apo-LapD and the trapped-inactive LapD (TMcys $S^{229}C/A^{602}C$), unconcentrated protein (0.5–3 mg/ml) was injected directly into the sample cell and

oscillated during data collection (10 × 1 s exposures). Data were collected using a dual Pilatus 100 K-S SAXS/WAXS detector and were background corrected and averaged using the program RAW. Only data showing no signs of radiation damage or aggregation (based on inspection of Guinier plots) were used for further analysis.

For the LapD•c-di-GMP complex and the LapD•c-di-GMP•LapG complex, samples were prepared identically to that for SEC-MALS. 100 µL of sample at ~10 mg/ml was injected onto the same Superdex 200 10/300 size-exclusion column pre-equilibrated with SEC-MALS buffer supplemented with 20 µM c-di-GMP, which was in-line with the SAXS sample cell. Each frame of scattering data was collected for two seconds, and 10–20 frames corresponding to a single protein peak were averaged and background corrected.

Buffer-subtracted scattering data were analyzed and scaled using the program Primus (*Konarev et al., 2003*). The program GNOM (*Svergun, 1992*) was used to determine the distance distribution functions P(r), radius of gyration ($R_g$), and maximum diameter of the particle ($D_{max}$). $D_{max}$ was adjusted so that distribution functions fell naturally to zero beyond $D_{max}$ and the best goodness-of-fit parameter was obtained. SAXS-based shape reconstructions were carried out with the programs DAMMIF (*Franke and Svergun, 2009*) and DAMMIN (*Svergun, 1999*). Briefly, 20 models were calculated by DAMMIF, and the program DAMAVER (*Volkov and Svergun, 2003*) was used to superimpose, filter and average these models. The resulting 'damstart' model from this DAMMIF analysis was used as an initial model for further refinement with DAMMIN to generate ten additional models. These ten DAMMIN models were superimposed, filtered and averaged using DAMAVER to produce the final model.

Simulated annealing and rigid-body fitting of individual structures of LapD domains to the scattering data were performed by the program SASREF (*Petoukhov and Svergun, 2005*) using an iterative protocol. This analysis was facilitated by the use of crystallographic models of individual soluble domains of LapD, including the periplasmic domain (PDB 4u64), the periplasmic domain bound to LapG (PDB 4u65), the autoinhibited S helix-GGDEF-EAL cytoplasmic module (PDB 3pjx) and the c-di-GMP-bound EAL domain dimer (PDB 3pju), as well as a homology model of LapD's HAMP domain (template PDB 2y21; (*Ferris et al., 2011*)). The detergent corona was built and refined using the program MEMPROT (*Pérez and Koutsioubas, 2015*). Structures were docked into the three-dimensional envelopes using SUPCOMB (*Kozin and Svergun, 2001*). Theoretical scattering curves were created and comparisons with experimental data made with CRYSOL (*Svergun et al., 1995*). To perform the SASREF analysis for the trapped-inactive form of LapD, models for the periplasmic, transmembrane, HAMP, and two autoinhibited domains were docked into the auto-inhibited LapD DAMMIN envelope, and a manually constructed micelle was placed around the transmembrane domain. SASREF was used to rotate and translate each individual domain to fit the scattering data, enforcing only restraints necessary to maintain contiguous peptide chains. The program MEMPROT was then used to generate and optimize a detergent micelle around the transmembrane domain of the resulting model. This model with the optimized MEMPROT-generated micelle was then re-run through SASREF in an identical manner to generate the final model for trapped-inactive LapD. For native, apo-LapD, the micelle generated from the trapped-inactive structure was used, and SASREF was run similarly except that the EAL domains of LapD were separated from their respective GGDEF domains to allow for flexibility. The identical procedure was performed for the LapD•c-di-GMP complex. For the LapD•c-di-GMP•LapG dimer-of-dimers, an additional restraint was enforced requiring close distance (3–6 Å) between the residues A[602] of the EAL domains, as experimentally determined and apparent in the c-di-GMP-bound EAL domain crystal structure.

For assessing EAL domain flexibility, the trapped-inactive LapD model generated by SASREF (described above) was used, and the residues connecting the EAL domain to the GGDEF domain of one LapD protomer were allowed to be remain flexible. The program Ensemble Optimization Method (EOM) (*Bernadó et al., 2007*; *Tria et al., 2015*) was used to generate 10,000 models in which this EAL domain was allowed to adopt any random orientation around the rest of the fixed LapD molecule. A second pool was generated in which this linker was allowed to be flexible in both protomers of LapD. By applying EOM (*Bernadó et al., 2007*), an ensemble of conformers was selected from each pool that best fit either the trapped-inactive, native, or LapD•c-di-GMP scattering data. Ensemble characteristics relative to the entire pool of models were assessed based on their respective $R_g$ and $D_{max}$. Inspection of individual models from the selected ensembles revealed

features that were consistent with the interpretation of the corresponding envelope and rigid-body models (data not shown).

Statistics associated with this and all other analyses are reported in *Figure 4—source data 1*. For all dimeric complexes, P1 symmetry was used based on better modeling statistics compared to modeling with P2 symmetry applied. For the dimer-of-dimers complex, data fit was only marginally better using P2 symmetry.

## Acknowledgements

We thank George O'Toole for feedback on the manuscript. Our work was supported by the NIH under grants R01-AI097307 (HS), T32-GM008500 (JPO), and F32-GM108440 (RBC). CHESS is supported by the NSF and NIH/NIGMS via NSF award DMR-1332208, and the MacCHESS resource is supported by NIH/NIGMS award GM-103485.

## Additional information

### Funding

| Funder | Grant reference number | Author |
|---|---|---|
| National Institute of Allergy and Infectious Diseases | R01-AI097307 | Holger Sondermann |
| National Institute of General Medical Sciences | F32-GM108440 | Richard B Cooley |
| National Institute of General Medical Sciences | T32-GM008500 | John P O'Donnell |

The funders had no role in study design, data collection and interpretation, or the decision to submit the work for publication.

### Author contributions

RBC, Conceptualization, Formal analysis, Funding acquisition, Investigation, Visualization, Writing—original draft; JPO'D, Conceptualization, Formal analysis, Investigation, Validation, Writing—original draft; HS, Conceptualization, Formal analysis, Supervision, Funding acquisition, Investigation, Visualization, Writing—original draft, Project administration

### Author ORCIDs

Holger Sondermann, http://orcid.org/0000-0003-2211-6234

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
