## [Decision Letter]

Thank you for submitting your article "Coincidence detection and bi-directional transmembrane signaling control a bacterial second messenger receptor" for consideration by *eLife*. Your article has been reviewed by three peer reviewers, and the evaluation has been overseen by a Reviewing Editor and Michael Marletta as the Senior Editor. The reviewers have opted to remain anonymous.

The reviewers have discussed the reviews with one another and the Reviewing Editor has drafted this decision to help you prepare a revised submission.

Summary:

The reviewers uniformly felt that the manuscript built on the authors' previous report, contained interesting data, and proposed a very interesting and plausible model. They also felt that the presentation needs to be greatly improved and that while the multiple techniques employed – size exclusion chromatography-multi-angle light scattering (SEC-MALS), cysteine- and photo-crosslinking, and small angle X-ray scattering (SAX) – provided much interesting data that supported the model, they did not definitively demonstrate its correctness. In addition, while the multiple techniques contributed to the strength of the article, they also made the logical flow of the manuscript difficult to follow and that the figures and figure legends were overly detailed and difficult to follow.

Essential revisions:

The main technical issues are:

SEC-MALS: What is the origin of the dn/dc value of 0.14 for LMNG detergent? This value is critical for determining the molecular weight estimates that are reported for the studies. The full reviews also contain detailed comments on the need to include c-di-GMP data, improvements in Figure 1, and the related discussion.

Cysteine- and photo- crosslinking: These studies need MS data to verify the crosslinks and must be provided. In addition, the monomeric state of the GFP needs to be explicitly addressed. Does the construct used have the A206K oligomerization-blocking mutation? The full reviews provide some guidance on improving the figures and discussion.

Small angle X-ray scattering (SAX): A direct comparison of distance distributions between LapD-Apo and LapD-c-di-GMP complex should be in panel B. Is Figure 4 properly labeled?

Reviewer #1:

This is a very interesting manuscript that builds on the authors' previous work on LapD and LapG's role in cyclic-di-GMP sensing and regulation of adhesins involved in biofilm formation. Here the authors show utilizing three different methods: SEC-MALS, site-specific crosslinking, and SAXS-based modeling, several states of LapD in response to interactions with LapG on its periplasmic side, and cyclic-di-GMP on its cytoplasmic side. This work is very well conducted and the manuscript is clearly written. I really don't see any further experimentation that is required. The fascinating part of this manuscript is that it shows the fluidity of all of these different states of LapD, and how this can fine tune and modify signal transduction.

If anything, it would be nice if the authors elaborated on their discussion a little more on how similar/different this system is to other cyclic-di-GMP signaling systems. They compare and contrast with the AI-2 system and the Tsr/Tar system, but not too much with other cyclic-di-GMP systems.

Reviewer #2:

In this manuscript, Cooley and Sondermann examine the regulation of the c-di-GMP receptor by its ligands. The authors employ a variety of techniques including SEC-MALS, cysteine crosslinking, photo crosslinking, and small angle X-ray scattering.

The manuscript investigates an interesting and important question, and the combination of a wide variety of biophysical techniques is a strength. The manuscript is difficult to follow at places, and the figures in particular are crowded and overly complex. In Figure 1 for instance, the inclusion of SEC traces in which c-di-GMP is absent from the running buffer does not seem relevant, since the ligand may simply be dissociating from the receptor during the experiment. With this caveat, the remainder of the SEC-MALS analysis is compelling.

Cysteine- and photo-crosslinking experiments provide a nice counterpart to the in vitro analyses, but again the presentation is difficult to follow given the very large amount of data inserted into one figure. While the data are generally consistent with the authors’ model, the potential for crosslinking to trap even rare conformations makes this approach less quantitative than the other analyses.

In the SAX experiments, a direct comparison of distance distributions between LapD-Apo and LapD-c-di-GMP complex would be more useful in panel B, since this state is an important component of the proposed model for an asymmetric dimer.

Overall the manuscript contains interesting data and proposes a plausible model for LapD activation, but suffers from overly complex figures and difficult logical flow. The model proposed by the authors is supported by the data presented, but not definitively demonstrated in my view. This model would be significantly strengthened by experimental measurement of cooperativity between c-di-GMP binding and binding of LapG, which is assessed only indirectly here. Specifically, what is the affinity of LapG to LapD in the absence of c-di-GMP, and to what extent is this changed upon the addition of c-di-GMP? Measurements by isothermal titration calorimetry would be even more compelling, as these would allow highly quantitative measurement of binding stoichiometry.

Technical points:

SEC-MALS experiments are analyzed using a dn/dc value of 0.14 for LMNG detergent. Where is this value from, and how was it determined? The accuracy of this value is important in obtaining reliable molecular weight estimates in SEC-MALS.

The superfolder GFP is described as monomeric. What does this mean? The originally reported superfolder GFP is not an obligate monomer, which could significantly compromise assessment of oligomerization of LapD. If the constructs carry the A206K oligomerization-blocking mutation this should be stated explicitly.

Reviewer #3:

This manuscript describes a new model for the activation of LapD. It is based on observations of the oligomerization state of the full-length detergent-solubilized LapD in the presence/absence of c-di-GMP and LapG using three techniques: (1) SEC-MALS, (2) Cys-cross-linking and (3) SAXS. The authors provide strong evidence that in the presence of LapG and c-di-GMP, LapD forms asymmetric tetramers, in which tetramerization is mediated by two of the four EAL domains, one from each LapD dimer. This data is very exciting and intriguing, providing a new model regarding LapD activation. While the experimental work is largely sound, the current work would be enhanced by providing a more clear mechanism that explains how this asymmetric tetramer is stabilized by LapG/c-di-GMP, since this is not obvious from the data presented (nor discussed in detail).

Experimental work.

1) MS data to verify the Cys-crosslinks (cutting out the bands from the gel/tryptic digest) would enhance the manuscript (and also allow one to determine what the 'grey' species are). This should be straightforward.

2) Are the SEC results identical for WT, TMcys and TMcys-mutants?

3) SAXS. Please use thinner lines for the figures/plots - the thick lines give the impression that the experimental and modeled data are more similar than they are. The envelope for the trapped-inactive state does not exhibit two-fold symmetry, although this is how it is modeled in the cartoon and discussed. Discussion about this discrepancy is warranted.

4) Major: The envelopes (or labels?) in Figure 4 and Figure 4—figure supplement 3 for the LapD (apo) and trapped-inactive LapD are switched and thus it is unclear which one is correct.

5) Major: The χ^2^ values for the manual docking figures are very high (poor fit). In contrast, the fits are improved by using quantitative fitting of models to the envelope. The fits in 4-3 are what should be shown in Figure 4, not the manual, more poorly fitting models. This would also alleviate the question about why there is so much density in the tetramer model in 4C that is unaccounted for.

---

## [Author Response]

*Essential revisions:*

*The main technical issues are:*

*SEC-MALS: What is the origin of the dn/dc value of 0.14 for LMNG detergent? This value is critical for determining the molecular weight estimates that are reported for the studies. The full reviews also contain detailed comments on the need to include c-di-GMP data, improvements in Figure 1, and the related discussion.*

Initially, we used this value as an estimate based on the published dn/dc of CHAPS (0.132), which is the predominant detergent in this experiment, and a detergent closely related structurally to LMNG, DDM (0.143). LMNG has the same lipid and maltose head groups as DDM but contains a quaternary carbon that joins two DDM groups.

However, in the revised manuscript, we have now determined all relevant dn/dc values experimentally following the approach described by Strop and Brunger (Protein Sci 2005, PMID: 16046633) (Figure 1—figure supplement 3). In particular, we determined dn/dc values for CHAPS, LMNG, and the LMNG/CHAPS (1:8) mixture that was used in the gel filtration buffer. In the process, we encountered a difference between the reported and measured dn/dc for CHAPS (0.132 vs 0.177). We determined dn/dc values for two other detergents used in the study by Strop and Brunger (Strop and Brunger, Protein Sci 2005, PMID: 16046633), LDAO and DDM, and the published values agree with our experimental values for these detergents. Hence, we relied on our experimentally determined dn/dc values for LMNG/CHAPS in the re-analysis of the light scattering data (Figure 1—figure supplement 3), and adjusted all molecular weight estimates in the revised manuscript accordingly. The Material and methods section was revised accordingly. The new dn/dc values had marginal effects on the protein molecular weights and hence did not impact our interpretation of the results.

Cysteine- and photo- crosslinking: These studies need MS data to verify the crosslinks and must be provided.

We have determined photocrosslinking specificity in the original publication on which this Research Advance is based upon. While we did not conduct mass spectroscopic analysis, we introduced non-natural amino acid residues at sites in the crystallographic LapD-LapG interface and also at positions chosen as negative controls that did not support crosslinking (Figure 6, Chatterjee et al., *eLife* 2014, PMID: 25182848). Furthermore, crosslinking relied on residue W^125^ in LapD, establishing unequivocally that this strategy reported on the crystallographic (and physiologically relevant) binding interface (Figure 9, Chatterjee et al., *eLife* 2014, PMID: 25182848 and also in this manuscript, Figure 2—figure supplement 1).

We agree that the crosslinking patterns in some samples are complex. However, we would like to point out that the main crosslinking products are specific and can be assigned unambiguously to specific cysteine positions based on available crystal structures and the numerous controls included in the experiment. For example, we demonstrated that cysteineless LapD-sfGFP shows no crosslinking bands under any conditions. Furthermore, we assigned specific crosslinking adducts to native transmembrane cysteine residues based on LapD variants with only a single cysteine residue. Based on the data presented, we can confidently and unambiguously assign bands marked with red, magenta, blue, and green symbols to the specific, indicated cysteine pairs, despite the complexity of the crosslinking patterns.

The only band/area in question for which we cannot assign the exact crosslinking position arises only from the combination of LapD^TMcys^ and A^602^C (gray asterisks). We argue that this minor ambiguity regarding only one of the main crosslinking products does not take away from the central message of the experiment, which allowed us to distinguish between different states by analyses of the crosslinking patterns. The ambiguous band is discussed as such, so we feel we did not overstate the analytical power of this experiment.

For the above reasons, we would like to point out that the suggested MS experiments would contribute very little new information. With our current experimental setup, MS experiments are not as straightforward as mentioned in the reviews for the following reason: The experiments shown here are based on in-gel fluorescence of partially purified protein. We chose this approach due to its incredible sensitivity, allowing us to work at very low, more physiologically relevant concentrations. To make the MS experiments work, we would have to scale up the protein preparations enabling us to work with a pure system. Next we would have to rely on appropriate peptide recovery and coverage of the cysteine-containing motifs. Considering the number of mutants used here, logistically and financially this is not a trivial investment for an analysis that provides no guarantees to produce the required data and more importantly, likely adds little new to the analysis. Finally, while the MS-based analysis was labeled as an essential revision, we note that only one reviewer made mention of this, and in doing so suggested this analysis would only “enhance” the manuscript rather than be critical to support the proposed model.

Given that all crosslinking data presented here are consistent with results from the two other, main experimental approaches (as noted by the reviewers and editor) and with previously determined crystal structures, we do not think MS analysis would provide additional essential support for the proposed model of LapD activation. However, we have extended the crosslinking Results section to clarify the rational behind the controls that led to the unambiguous determination of sites supporting crosslinking in the various states.

*In addition, the monomeric state of the GFP needs to be explicitly addressed. Does the construct used have the A206K oligomerization-blocking mutation? The full reviews provide some guidance on improving the figures and discussion.*

All assays were performed with monomeric superfolder GFP (sfGFP) with a V^206^K mutation that blocks oligomerization tendencies of the parent sfGFP, which we now state explicitly in the Material and methods section. Competition experiments with dark (non-sfGFP-tagged) LapD also rule out any effect due to sfGFP dimerization.

*Small angle X-ray scattering (SAX): A direct comparison of distance distributions between LapD-Apo and LapD-c-di-GMP complex should be in panel B. Is Figure 4 properly labeled?*

We would like to note that this comparison is shown in Figure 4, right panel. Indeed, an error was made in the labeling of Figure 4—figure supplement 3; the envelope and docking models for trapped-inactive and apo-LapD were accidentally swapped. We thank the reviewers for catching this mistake, which we corrected in the revised Figure 4—figure supplement 3.

*Reviewer #1:*

*This is a very interesting manuscript that builds on the authors' previous work on LapD and LapG's role in cyclic-di-GMP sensing and regulation of adhesins involved in biofilm formation. Here the authors show utilizing three different methods: SEC-MALS, site-specific crosslinking, and SAXS-based modeling, several states of LapD in response to interactions with LapG on its periplasmic side, and cyclic-di-GMP on its cytoplasmic side. This work is very well conducted and the manuscript is clearly written. I really don't see any further experimentation that is required. The fascinating part of this manuscript is that it shows the fluidity of all of these different states of LapD, and how this can fine tune and modify signal transduction.*

*If anything, it would be nice if the authors elaborated on their discussion a little more on how similar/different this system is to other cyclic-di-GMP signaling systems. They compare and contrast with the AI-2 system and the Tsr/Tar system, but not too much with other cyclic-di-GMP systems.*

We expanded discussions regarding the relevance of our results for other c-di-GMP signaling systems. However, the dearth of structural data and mechanistic insight for most of these systems hampers direct comparisons with the Lap system – an aspect that we also discuss in the revised manuscript.

*Reviewer #2:*

*In this manuscript, Cooley and Sondermann examine the regulation of the c-di-GMP receptor by its ligands. The authors employ a variety of techniques including SEC-MALS, cysteine crosslinking, photo crosslinking, and small angle X-ray scattering.*

*The manuscript investigates an interesting and important question, and the combination of a wide variety of biophysical techniques is a strength. The manuscript is difficult to follow at places, and the figures in particular are crowded and overly complex. In Figure 1 for instance, the inclusion of SEC traces in which c-di-GMP is absent from the running buffer does not seem relevant, since the ligand may simply be dissociating from the receptor during the experiment. With this caveat, the remainder of the SEC-MALS analysis is compelling.*

In direct response to this comment, we removed the chromatogram and SEC-MALS analysis of LapD-LapG-c-di-GMP without c-di-GMP in the mobile phase from Figure 1. However, we kept them as a Figure Supplemental to Figure 1 and as part of the manuscript since they make exactly the point the reviewer indicates, that the c-di-GMP-bound state is not stable in solutions lacking c-di-GMP, which we discuss in the manuscript.

*Cysteine- and photo-crosslinking experiments provide a nice counterpart to the* in vitro *analyses, but again the presentation is difficult to follow given the very large amount of data inserted into one figure. While the data are generally consistent with the authors’ model, the potential for crosslinking to trap even rare conformations makes this approach less quantitative than the other analyses.*

We agree with above statement that the crosslinking approach is not quantitative, which is why we presented the primary, qualitative data. Nevertheless, with all controls provided in the original manuscript, we can determine crosslinking-supporting positions for all but one band. More importantly, the crosslinking patterns support the point we discuss in the manuscript, which is that there are multiple, clearly distinguishable states of apo, c-di-GMP-bound, and c-di-GMP/LapG-bound LapD.

While the data cannot be quantified in absolute terms based on our data, the relative crosslinking efficiencies and banding patterns paint a clear picture. Combined with our previously published structural data, the provided controls, the overall robustness of the crosslinking approach, and the consistency across the three independent methods of characterization presented here (SEC-MALS, crosslinking, and SAXS), it is very hard to envision a scenario in which the dominant crosslinking bands we discuss are due to rare or poorly populated conformations. Based on our cumulative data, we are confident that this experimental approach actually reports on the main interactions identified in crystal structures.

*In the SAX experiments, a direct comparison of distance distributions between LapD-Apo and LapD-c-di-GMP complex would be more useful in panel B, since this state is an important component of the proposed model for an asymmetric dimer.*

We would like to point out that this comparison is shown in Figure 4, right panel.

*Overall the manuscript contains interesting data and proposes a plausible model for LapD activation, but suffers from overly complex figures and difficult logical flow. The model proposed by the authors is supported by the data presented, but not definitively demonstrated in my view. This model would be significantly strengthened by experimental measurement of cooperativity between c-di-GMP binding and binding of LapG, which is assessed only indirectly here. Specifically, what is the affinity of LapG to LapD in the absence of c-di-GMP, and to what extent is this changed upon the addition of c-di-GMP? Measurements by isothermal titration calorimetry would be even more compelling, as these would allow highly quantitative measurement of binding stoichiometry.*

We have edited text and figures to improve clarity. With regard to cooperativity, the suggested experiment was documented in our previous publication based on photocrosslinking data (Figure 7C, Chatterjee et al., *eLife* 2014, PMID: 25182848), which we agree is important and indeed motivated the mechanistic studies in this Research Advance. We considered isothermal titration calorimetry as an approach to measure these interactions but concluded that this technique is less than ideal for this system. There are simply too many molecular events to deconvolute that occur during activation of LapD (including c-di-GMP binding to LapD, LapG binding to both apo- and c-di-GMP bound LapD, EAL domain dimerization, and disengagement of the EAL domain from the S helix, all of which will contribute to the overall energetics of the system). This complexity renders a direct comparison by isothermal titration calorimetry of LapD binding to LapG in the absence and presence of c-di-GMP challenging. However, we developed and included in the revised manuscript a quantitative binding assay based on fluorescence anisotropy, and report the affinity values the reviewer requested (Figure 1—figure supplement 2).

*Technical points:*

*SEC-MALS experiments are analyzed using a dn/dc value of 0.14 for LMNG detergent. Where is this value from, and how was it determined? The accuracy of this value is important in obtaining reliable molecular weight estimates in SEC-MALS.*

See above. In the revised manuscript, we have experimentally determined the relevant dn/dc value for LMNG, CHAPS, and the LMNG/CHAPS mixture, and re-analyzed all molecular weight estimates using accurate measurements.

*The superfolder GFP is described as monomeric. What does this mean? The originally reported superfolder GFP is not an obligate monomer, which could significantly compromise assessment of oligomerization of LapD. If the constructs carry the A206K oligomerization-blocking mutation this should be stated explicitly.*

In superfolder GFP, the dimer-supporting residue is V^206^, and we used a V^206^K mutant in all our assays. This experimental detail is more clearly stated in the revised manuscript (see Material and methods section).

*Reviewer #3:*

*This manuscript describes a new model for the activation of LapD. It is based on observations of the oligomerization state of the full-length detergent-solubilized LapD in the presence/absence of c-di-GMP and LapG using three techniques: (1) SEC-MALS, (2) Cys-cross-linking and (3) SAXS. The authors provide strong evidence that in the presence of LapG and c-di-GMP, LapD forms asymmetric tetramers, in which tetramerization is mediated by two of the four EAL domains, one from each LapD dimer. This data is very exciting and intriguing, providing a new model regarding LapD activation. While the experimental work is largely sound, the current work would be enhanced by providing a more clear mechanism that explains how this asymmetric tetramer is stabilized by LapG/c-di-GMP, since this is not obvious from the data presented (nor discussed in detail).*

This is an obvious next question we would like to understand better. However, based on our current structural and mechanistic understanding, any further discussion would be highly speculative.

*Experimental work.*

*1) MS data to verify the Cys-crosslinks (cutting out the bands from the gel/tryptic digest) would enhance the manuscript (and also allow one to determine what the 'grey' species are). This should be straightforward.*

This is not a trivial experiment for several reasons (see above). In a purified system, it may be straightforward to cut out the bands in a gel and subject them to mass spectrometry. However, the assays shown here rely on partially purified membranes making use of sfGFP-fusion proteins for detection. The background of non-fluorescent proteins is therefore too high to enable straightforward mapping of disulfide bonds by mass spectroscopy.

More importantly, we confidentially assigned all but one dominant crosslinking products to specific cysteine pairs based on exhaustive controls. The only band/area that we cannot assign with high confidence is labeled (grey markings) and discussed appropriately. However, this minor caveat does not take away from the overall conclusions drawn from this and all other experiments shown here. While we agree such analyses may provide additional proof of the sites of crosslinking, we do not consider it an essential control in light of all the data we included already.

*2) Are the SEC results identical for WT, TMcys and TMcys-mutants?*

SEC profiles and SEC-MALS analysis are comparable for these proteins. SEC-MALS data for the TMcys variant, a representative mutant (TMcys-A^602^C), and a LapD-sfGFP fusion protein are now included in the revised manuscript (Figure 2—figure supplement 1).

*3) SAXS. Please use thinner lines for the figures/plots - the thick lines give the impression that the experimental and modeled data are more similar than they are. The envelope for the trapped-inactive state does not exhibit two-fold symmetry, although this is how it is modeled in the cartoon and discussed. Discussion about this discrepancy is warranted.*

We have made the suggested changes and added more discussion regarding the asymmetry of the apparent trapped dimer. Specifically, we added the following sentence to the Results section: “Even the trapped-inactive LapD dimer could not be modeled with two-fold symmetry indicating that crosslinking stabilizes or introduces some sort of a strained conformation (Figure 4).”

*4) Major: The envelopes (or labels?) in Figure 4 and Figure 4—figure supplement 3 for the LapD (apo) and trapped-inactive LapD are switched and thus it is unclear which one is correct.*

During a re-organization of Figure 4—figure supplement 3, the panels of the structural models for the two states were switched by accident. We corrected this mistake in the revised manuscript. Figure 4 showed the correct envelopes/labeling.

*5) Major: The χ^2^ values for the manual docking figures are very high (poor fit). In contrast, the fits are improved by using quantitative fitting of models to the envelope. The fits in 4-3 are what should be shown in Figure 4, not the manual, more poorly fitting models. This would also alleviate the question about why there is so much density in the tetramer model in 4C that is unaccounted for.*

We considered the suggested change, however decided to keep the current organization. We were careful in our Discussion not to overstate the accuracy of manual fits by stating explicitly the goodness of fits in the Figure Supplement. The manually docked models were compared to the envelope reconstructions predominantly with regard to their overall dimensions and shapes. Unlike the computationally fit models, which included more degrees of freedom, the manually docked models are based on published and functionally validated crystal structures.

Considering that we report comparisons accurately, it becomes a question of preference whether to show in the main figure models that preserve physiologically important conformations observed in crystal structures despite poorer fits due to the inherent constraints, or to show models with better fits but potentially disrupted interfaces due to the higher degrees of freedom allowed during the modeling. At the resolution of SAXS-based modeling and the fact that purified LapD is a multi-domain, micelle-containing protein, some ambiguity in the data can be expected. The unaccounted for density in the tetramer model coincides with the transmembrane/micelle region, which is likely the area of highest ambiguity with regard to the accuracy of the modeling process. As a result of these careful considerations, we decided to include several approaches and focus our discussion on the consensus interpretation emerging from this cross-validation.